# Safety and Efficacy of a First-Line Chemotherapy Tailored by G8 Score in Elderly Metastatic or Locally Advanced Gastric and Gastro-Esophageal Cancer Patients: A Real-World Analysis

**DOI:** 10.3390/geriatrics7050107

**Published:** 2022-09-29

**Authors:** Ina Valeria Zurlo, Carmelo Pozzo, Antonia Strippoli, Samantha Mignogna, Michele Basso, Raffaella Vivolo, Giovanni Trovato, Michele Ciaburri, Franco Morelli, Emilio Bria, Silvana Leo, Giampaolo Tortora

**Affiliations:** 1Medical Oncology Unit, “Vito Fazzi” Hospital, 73100 Lecce, Italy; 2Medical Oncology Unit, Comprehensive Cancer Center, Fondazione Policlinico Universitario “A. Gemelli” IRCCS, 00168 Rome, Italy; 3Medical Oncology Unit, Gemelli Molise, 86100 Campobasso, Italy; 4Geriatric Unit, Fondazione Policlinico Universitario “A. Gemelli” IRCCS, 00168 Rome, Italy

**Keywords:** gastric cancer, elderly, toxicity, G8 score, comprehensive geriatric assessment

## Abstract

Introduction: Gastric (GC) and gastro-esophageal cancer (GEC) are common neoplasms in the elderly. However, in clinical practice, the correct strategy for elderly patients who might benefit from chemotherapy (CT) is unknown. Prospective data are still poor. In this context, we performed a retrospective analysis of GC patients aged ≥75 years and treated at our institutions. Material and Methods: We retrospectively analyzed 90 patients with confirmed metastatic GC or GEC, treated with an upfront CT. Inclusion criteria were patients aged ≥75 years, PS 0–2, normal bone marrow/liver/renal function and no major comorbidities. All patients received a G8 score, and some patients with G8 ≤14 received a comprehensive geriatric assessment (CGA). The primary goal was to perform a safety evaluation based on the incidence of adverse events (AE), and the secondary goal was to determine the efficacy (PFS and OS). The chi-square test and the Kaplan–Meier method were used to estimate the outcomes. The statistical significance level was set at *p* < 0.05. Results: Toxicity rates were quite low: G1/G2 (51.1%) and G3/G4 (25.5%). No toxic deaths were reported. The median PFS was 6.21 months and the median OS 11 months. The G8 score and PS ECOG significantly influenced both PFS and OS. A statistically significant correlation among G8, weight loss, hypoalbuminemia and risk of G3/G4 adverse events was also found. Conclusion: Our research on selected elderly patients did not detect broad differences of efficacy and tolerability compared to a young population. Our study, although retrospective and small-sized, showed that G8 score might be an accurate tool to identify elderly GC/GEC patients who could be safely treated with CT, further recognizing patients who could receive a doublet CT and who may require a single agent chemotherapy or a baseline dose reduction.

## 1. Introduction

Gastric cancer (GC) is the third most common cause of cancer-related death worldwide, and its incidence is increasing. It is particularly prevalent in individuals aged between 50 and 70 years [1,2]. Despite the growth of the elderly population, the management of elderly oncological patients is not effective. The greatest difficulty is related to the frailty of this population due to physiological changes in organ function and increased risk of complications.

In real world clinical practice, considering the higher toxicity rate risk associated with a doublet or triplet-chemotherapy strategy, either oral monotherapy or combined schedules with reduced doses are commonly planned for elderly patients. Elderly and young GC patients differ in various respects, not least in terms of the prevalence of comorbidities and functional disorders [3]. Furthermore, elderly patients make up a heterogenous population, with some showing states of health similar to a younger age group and others presenting with cardiovascular, respiratory, mental, renal or metabolic disease [4].

Currently, elderly GC patients are under-represented in randomized clinical trials. As a consequence, international guidelines are mainly based the results obtained on younger population [5,6,7]. Therefore, the right strategy for fit or frail elderly patients remains unclear, and defining the correct and most effective treatment to ensure a satisfactory quality of life is an especially tricky task [8,9,10].

Some evidence has demonstrated that capecitabine monotherapy or an oxaliplatin or cisplatin containing-regimen is well tolerated in patients >70 years of age, with similar outcomes to those in a younger cohort [11,12,13,14].

Some subgroup analyses have underlined that selected elderly patients could obtain a similar survival benefit from chemotherapy compared to a younger group [11,12,13,14]. Nevertheless, this hypothesis should be considered with caution, as few elderly patients are included in prospective clinical trials, and retrospective studies do not yield exhaustive and definitive results.

The International Society of Geriatric Oncology recommends that elderly cancer patients undergo a Comprehensive Geriatric Assessment (CGA) to detect problems which are not readily identified by routine physical examinations or medical history in order to predict cancer treatment-related toxicities, to predict survival, and to assist in cancer treatment decisions. Geriatric Assessment (GA) is a multidimensional and interdisciplinary evaluation tool that makes it possible to identify functional, nutritional, cognitive, psychological, social support, and comorbidity factors [15]. Although GA is valuable in oncology, a full GA is time-consuming. Geriatric screening tools such as G8 are recommended to identify patients in need of further evaluation via a full CGA [16,17,18,19,20].

The G8 screening tool consists of seven items dealing with food intake, weight loss, mobility, neuropsychological problem, body mass index, prescription drug, and self-perception of health. It is based on the Mini-Nutritional Assessment (MNA) questionnaire and was developed specifically for elderly cancer patients [18]. G8 takes 3 ± 5 min. Scores range from 0 (poor) to 17 (good); a score of 14 is considered abnormal.

The test can predict survival and identify frailty and patients with an increased chemotherapy (CT) toxicity or adverse events risk [16,17]. However, the CGA is not an easy instrument, requiring time and resources. In most centers, close collaboration between oncologists and geriatricians is often an unmet need. In response to this critical issue, a shorter and easier frailty assessment tool, the geriatric 8 (G8) questionnaire, was developed [18].

However, the management of the elderly is complicated due to the lack of randomized controlled trials which include this subgroup.

On these bases, we performed a retrospective analysis to evaluate the tolerability of antitumoral treatments by GC and GEC elderly population (≥75 years old) treated at our institution, exploring the safety and efficacy of first-line chemotherapy and analyzing the clinicopathological features that might guide clinical strategy choices.

## 2. Material and Methods

This retrospective analysis included elderly patients ≥75 years of age with histologically proven GC or GEC treated at the oncology units of Fondazione Policlinico Universitario “A. Gemelli” IRCSS in Rome, Fondazione “A. Gemelli” Molise in Campobasso and Ospedale “Vito-Fazzi” in Lecce between January 2013 and June 2021. Inclusion criteria were: (a) histologically confirmed adenocarcinoma of the stomach or gastro-esophageal junction with metastatic or locally advanced disease; (b) patients ≥75 years of age that had received CT for a metastatic disease; (c) known concomitant illnesses and polypharmacy; (d) known hepatic, renal and bone marrow function, (e) a left ventricular ejection fraction (LVEF) of ≥50%; (f) a performance status (PS) of 0–2, according to the Eastern Cooperative Oncology Group; and (g) complete information regarding height, weight, weight loss in the last three months, treatment outcomes and toxicities.

All patients underwent G8 screening [18], and those with G8 scores of ≤14 also received a CGA, which evaluates functional status, comorbidities, cognition, psychological status, social functioning and nutritional status, as measured through the activities of daily living (ADL) and instrumental activities of daily living (IADL) scales, as well as by the geriatric depression and mini-mental state scales [21,22,23,24,25,26]. Comorbidities were rated according to the Cumulative Illness Rating Score for Geriatricians (CIRS-G) [27]. Performance status (PS) was evaluated according to European Cancer Organization group (PS ECOG).

This study was performed in accordance with the rules of the local Ethics Committee and the Declaration of Helsinki. All patients provided a written consent for use of their clinical data.

Demographic and clinicopathological characteristics were analyzed using the chi-square test.

The objective of the study was to evaluate the tolerability and adverse events (AEs) according to the Common Terminology Criteria for Adverse Events (CTCAE) v4.0.

The secondary goal was to evaluate the efficacy in terms of PFS and OS.

The Kaplan-Meier method was used to estimate OS and PFS, and a Cox regression model was employed to estimate hazard ratios (HR) and two-sided 95% confidence intervals (CI).

PFS was defined as the time from the beginning of treatment until the date of clinical/radiological progression or death, whichever occurred first, or that of last follow-up visit (censored). OS was defined as the time from treatment initiation until the date of death from any cause or last follow-up visit (censored).

The whole metastatic population was divided into two groups according to G8 score (normal >14 and abnormal ≤14), and a comparison in terms of survival outcome (PFS and OS) between the two cohorts was performed. Patients received doublet-CT or a single agent regimen, defined as mono-CT. Trastuzumab was not counted in the number of drugs.

Then, a univariate analysis was performed to correlate survival endpoints with other clinical factors, i.e., age (> vs. <80 years old), G8 (≤14 vs. >14), tumor location (proximal vs. distal), BMI (< vs. >20 kg/m^2^), weight loss (> vs. <10%), PS (0–1 vs. 2), number of comorbidities (<3 vs. >3), polypharmacy (<3 vs. >3 drugs taken), chemotherapy (mono-CT vs. doublet-CT) and albumin count (>3.5 g/dL vs. >3.5 g/dL). Finally, statistically significant clinical variables were tested. The statistical significance level was set at *p* < 0.05. Data were analyzed using MedCal Statistical software (MedCalc version 20.115, European Customers). 

## 3. Results

The records of 700 patients with metastatic or locally advanced GC and GEC were reviewed. According to the inclusion criteria, only 90 patients were eligible: 72 (80%) with confirmed GC and 18 (20%) with GEC. Ten patients (11.1%) had locally advanced disease while the other 80 (88.9%) had metastatic disease. Median age was 78 years (range 75–87) and twenty patients (22.2%) were older than 80 years. Twenty-eight (31.1%) were females and 62 (68.9%) males; 38 (42.2%) had a PS 2 and 52 (57.7%) a PS 0–1. All patients underwent echocardiographic assessment (LVEF > 50%) and G8 testing. Sixty-six patients (73.3%) had an abnormal G8 score (≤14) while 24 patients (26.7%) had a normal score. Only ten patients with abnormal G8 scores (<12) underwent CGA; all patients with G8 score ≤14 received an immediate CT dose reduction, and supportive care, overseen by a palliative simultaneous care assistant was planned. Patients had a median number of comorbidities of 2 (range 0–6); the most frequent were hypertension (45/50%), diabetes (25/27.7%), atherosclerosis (23/25.5%), osteoporosis (20/22.2%), hypercholesterolemia (10/11.1%), benign prostatic hyperplasia (25/27.7%), and chronic obstructive pulmonary disease (COPD) (18/20%). Two patients also had neurological disease (Parkinson and Alzheimer’s disease). The median number of drugs taken at baseline before starting CT was 3 (range 0–8). A median weight loss of 6 kg (range 0–18) at diagnosis was detected. Twenty-two patients (24.4%) were underweight, 5 (5.6%) were overweight, and the remaining 63 (70%) were of normal weight. The principles sites of metastasis were liver (45; 50%), peritoneum (30; 33.3%), lung (8; 8.88%), lymph-nodes (35; 38.8%), and bone (3; 3.3%).

Fourteen patients (15.6%) were receiving a combination of Cisplatin/5-Fluoruracil/Trastuzumab, according to HER2-hyperepression; 50 patients (55.5%) were receiving a doublet combination of cisplatin or oxaliplatin (6/44) combined with capecitabine/5-fluorouracil (6/44); and 26 patients (28.9%) were receiving fluoropyrimidines as a single agent. The median number of administered cycles was seven (range 1–17). Twenty-two patients with G8 >14 received Trastuzumab-based CT according Her2-expression, while 2 patients received a single-agent CT due to comorbidities. Patients with G8 scores ≤14 underwent doublet (42; 46.6%) or single-agent CT (24; 26.6%) due to comorbidities and PS ECOG. At progression, twenty-seven patients (30%) also received a second-line regimen and five a third-line CT. Baseline characteristics are reported in Table 1.

## 4. Safety Outcome: Adverse Events

Adverse events occurred in 69 patients (65.8%): G3/G4 and G1/2 in 28 (31.1%) and 46 (51.1%) patients, respectively (Table 2). Hematological G3/4 toxicity included neutropenia (12, 13.3%), anemia (2, 2.2%), and thrombocytopenia (4, 4.4%). No febrile neutropenia was reported.

Nonhematological G3/4 toxicity included mucositis (9, 10%), diarrhea (7, 7.8%), neuropathy (5, 5.5%), anorexia (4; 4.4%), and asthenia (15, 16.6%). The most frequent G1/2 toxicity included anemia (12, 13.3%), asthenia (23, 25.5%), mucositis (8, 8.8%), diarrhea (10, 11.1%), neutropenia (9, 10%), thrombocytopenia (6, 6.6%), anorexia (11; 12.2%), dysgeusia (12, 13.3%), and neuropathy (8, 8.8%). Nausea and vomiting were infrequent (G1 in 9 and 7 patients; 10% and 7.7%). No toxic deaths were reported.

Only ten patients (11.1%) received full doses of chemotherapy at baseline, while for the other 80 patients (88.9%), the treatment was started at reduced doses. Nevertheless, the treatment was further reduced, delayed, or interrupted for 23 (25.5%), 53 (58.8%), and 7 patients (7.7%), respectively. Nine patients were also supported by G-CSF, and two patients were hospitalized.

## 5. Efficacy Outcome

The median PFS was 6.21 months (95% CI 5.6–7.13) and the median OS 11 months (95% CI 10.64–13.85) (Figure 1).

Based on our univariate analysis, weight loss, G8 score, PS ECOG, and chemotherapy schedules (mono-CT vs. doublet-CT) were correlated with PFS, whereas G8 score, PS ECOG, and type of chemotherapy were correlated with OS (Table 2). The multivariate analysis G8 score and PS ECOG influenced both PFS and OS (*p* = 0.0012 and *p* ≤ 0.0001 for PFS and *p* = 0.036 and *p* < 0.0001 for OS) (Table 3).

A correlation between G8 score and albumin count (*p* = 0.0113), weight-loss (<10%) (*p* = 0.0288), G3/G4 toxicities (*p* = 0.0148), type of chemotherapy (mono-CT vs. doublet-CT) (*p* = 0.0197), and PS ECOG (*p* = 0.0002) was found, as described in Table 4. No correlation between G8 score and BMI, liver or peritoneum metastasis or tumor sidedness was reported.

## 6. Conclusions

Despite the growing elderly population, in clinical practice, this demographic often tends to get overlooked, receiving less aggressive surgery and less intensive CT. The hesitation to recommend systemic CT in this population is related to comorbidities, pharmacokinetics, and pharmacodynamics age-related changes, which may lead to a higher toxicity. There is also a lack of prospective studies on this population.

Geriatric assessments based on nutritional status and comorbidities might help to identify elderly patients who could benefit from CT, allowing clinicians to plan the best therapy. Thus, determining the optimal treatment strategy for elderly patients requires close collaboration with geriatricians.

Our retrospective analysis suggested that G8, the most widely used instrument to define the heterogeneity and vulnerability of elderly, might be an appropriate tool to guide patient selection and estimate survival outcome. However, the CGA might be considered more accurate than G8 and could be incorporated in clinical oncology practice. Close cooperation between geriatricians and oncologists might help clinicians identify CT risks, weigh the benefits, and predict morbidity and mortality [21].

In our report, CGA was performed in only ten patients with an abnormal G8 score (vs. 66) due to the absence of a personalized geriatrician for most of the study period. The complexity and vulnerability of this group due to age, polypharmacy, and poor GC prognosis highlight the high need for early simultaneous care [28].

In our clinical practice, G8 and CT reduction or delay have been shown to facilitate the proper use of CT among 80-year-old patients with acceptable toxicity in patients with poor prognoses. Patients aged over 80 with Her2-hyperepressed GC received a platinum-based regimen with a baseline dose reduction without serious AE.

Our findings support the need of a suitable selection process and collaboration with geriatricians to ensure that tailored treatments are applied to achieve toxicity rates similar to those seen in the younger population, to maintain a reasonable risk-benefit ratio, and to improve survival outcomes in this underrepresented population.

In our analysis, parameters such as G8 score and PS ECOG were the main prognostic factors. A correlation between G8 and some clinical features related to malnutrition (weight loss, hypoalbuminemia) was found. Others prospective studies and surgical evidence have demonstrated a correlation between albumin count, malnutrition, and survivals [29,30]. A weight loss >10% influenced PFS at univariate analysis, suggesting the need of nutritional status evaluations of elderly patients with metastatic gastric cancer. Malnutrition is common in elderly patients affected by age-related conditions (dementia, oral and dental disorders, malabsorption), increasing the risk of malnutrition and potentially influencing the mortality of cancer patients [31,32,33]. Therefore, malnutrition should be detected using a validated nutritional screening tool, e.g., the Mini-Nutritional Assessment (MNA), which is based on an easy and cost-effective measurement of serum albumin count, in order to identify at-risk patients.

In real-life practice, the choice of strategy for elderly patients is based on physical condition, cognitive function, the presence of frailty or comorbidities, and laboratory data; however, it is difficult to achieve a good tolerability drugs while maintaining quality of life. In our study, 42 patients received a doublet combination despite a G8 score <14, showing a correlation between G8 score and a significant risk of G3/G4 toxicity (*p* = 0.0148).

Our study confirmed that GC or GEC patients with poor PS ECOG derived minimal or no benefit from systemic CT, underlying the importance of patient selection considering clinical condition and nutrition. Despite the selection of a >75-year-old population, our study revealed similar toxicities and survival rates to those reported in other studies [34], without toxic death, indicating that the elderly could tolerate CT if a dose reduction and a tailored CT regimen according G8 score and PS ECOG are planned [35].

Our study had some limitations. Firstly, the sample size and the retrospective analysis limited the power of our results and may have introduced selection bias. Although we have treated 700 patients over 7 years, only 90 patients affected by advanced GC or GEC were included in our analysis, which shows that in clinical practice, few elderly patients received CT.

Our study, although retrospective and small, confirms how G8 score and geriatricians collaboration are needed, helping oncologists with strategy selection (mono-CT versus combined-regimen), baseline reduction dose, and nutritional or supportive care, allowing tailored treatments to be created for a frail population without increasing toxicity and preserving quality of life. This is particularly crucial for advanced GC and GEC, which are commonly associated with rapid PS deterioration and very limited efficacy of available treatments, not only in the elderly but in patients of all age categories.

Published studies using CGA-intervention for patients with cancer are still rare. The spread of CGA might serve to encourage the development of novel clinical trials which could provide information about the impact of CT on physical function, survival, and functional independence, in addition to traditional survival endpoints [36]. Considering the severity and aggressiveness of advanced GC and GEC in elderly patients, prospective trials are needed to distinguish between patients that might benefit from CT and those for whom supportive care is the best choice.

## Figures and Tables

**Figure 1 geriatrics-07-00107-f001:**
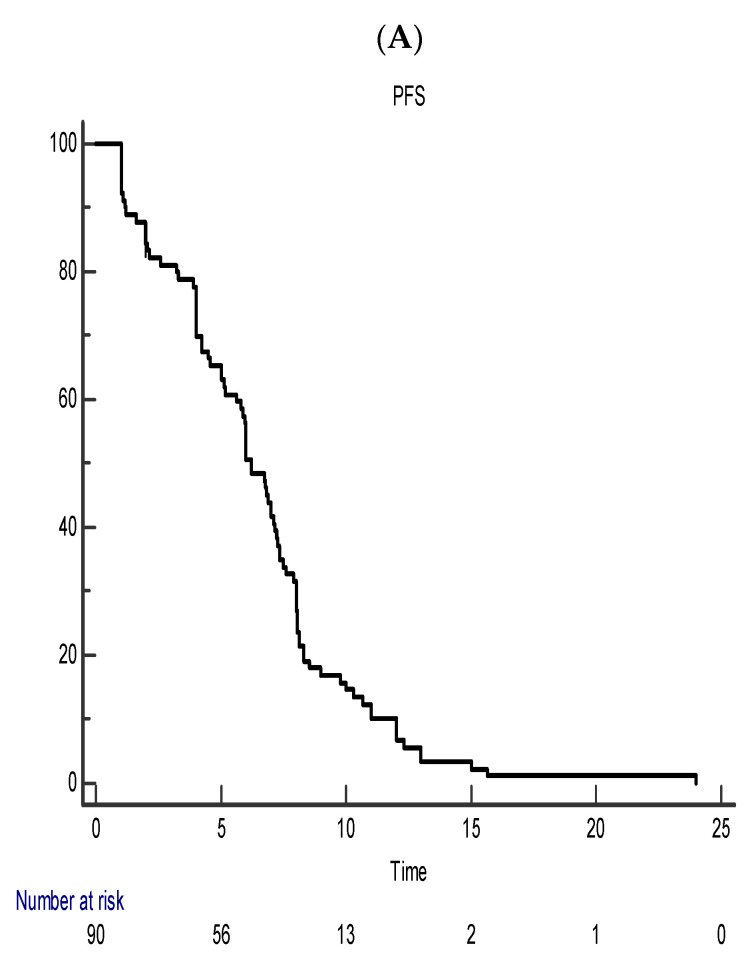
(**A**,**B**). Median PFS and OS in the whole metastatic population.

**Table 1 geriatrics-07-00107-t001:** Patient characteristics.

	Overall N (%)
**Characteristic**	90 (100)
**Age**	
Median (years) Range (years) Elderly (≥80)	78 75–87 20 (22.2)
**Sex**	
Male Female	62 (68.9) 28 (31.1)
**Comorbidities**	
Hypertension Diabetes Atherosclerosis Osteoporosis Hypercholesterolemia Benign prostatic hyperplasia COPD	45 (50) 25 (27.7) 23 (25.5) 20 (22.2) 10 (11.1) 25 (27.7) 18 (20)
**Sidedness**	
GC GEC	72 (80) 18 (20)
**Locally Advanced** **Metastatic**	10 (11.1) 80 (88.9)
**ECOG-PS**	
0–1 2	52 (57.8) 38 (42.2)
**G8 score**	
≤14 >15	66 (73.3) 24 (26.7)
**Median weight loss**	6 kg (0–18)
**Median number of comorbidities** Range	2 0–6
**Median number of drugs taken** Range	3 0–8
**Most frequent comorbidities**	
Hypertension Diabetes Atherosclerosis Osteoporosis Benign prostatic hyperplasia COPD	45 (50) 25 (27.7) 23 (25.5) 20 (22–2) 25 (27.7) 18 (20)
**BM1**	
<19 20–25 >25	22 (24.4) 63 (70) 5 (5.6)
** *CT* **	
Trastuzumab-based CT Doublet-CT Mono-CT	14 (15.5) 50 (55.5) 26 (28.8)
**Median cycles administered** Range	7 1–17
**Second-line chemotherapy** **Third-line chemotherapy**	15 (21.4) 5 (5.5)

**Table 2 geriatrics-07-00107-t002:** Hematological and nonhematological AE according CTCAE v.4.

	Overall N (90)			
Adverse Events (AE)	G-G2 N (%) 46 (51.1)	Mono/Doublet-CT	G3-G4 N (%) 28 (31.1)	Mono/Doublet-CT
**Hematologic AE**				
Neutropenia	9 (10)	5/9	12 (13.3)	5/7
Anemia	12 (13.3)	3/8	2 (2.2)	0/2
Thrombocytopenia	6 (6.6)	3/3	4 (4.4)	3/4
**Non-hematological AE**				
Asthenia	23 (25.5)	6/11	15 (16.6)	3/5
Diarrhea	10 (11.1)	4/6	7 (7.8)	8/0
Mucositis/stomatitis	8 (8.8)	2/6	9 (10)	1/2
Neuropathy	8 (8.8)	0/5	5 (5.5)	0/5
Anorexia	11 (12.2)	2/9	4 (4.4)	1/3
Dysgeusia	12 (13.3)	1/11	-	-
Nausea	9 (10)	3/6	-	-
Vomiting	7 (7.7)	1/6		

**Table 3 geriatrics-07-00107-t003:** Univariate and multivariate analysis.

	Univariate Analysis	
	PFS	OS
**Variable**	**HR (95% CI); *p*-Value**	**HR (95% CI); *p*-Value**
**Albumin count**		
<3.5 vs. >3.5 g/dL	1.3 (0.79–2.20); *p* = 0.234	1.12 (0.66–1.88); *p* = 0.64
**BMI**		
<20 vs. >20 kg/m^2^	1.10 (0.67–1.81); *p* = 0.67	0.94 (0.57–1.56); *p* = 0.82
**Weight loss**		
<10 vs. >10%	1.93 (1.26–2.96); *p* = 0.0008	1.47 (0.95–2.28); *p* = 0.06
**Gender**		
Female vs. Male	0.88 (0.55–1.40); *p* = 0.57	0.89 (0.55–1.45); *p* = 0.64
**PS ECOG**		
0–1 vs. 2	0.31 (0.17–0.51); *p <* 0.0001	0.28 (0.16–0.49); *p* < 0.0001
**G8 score**		
Normal vs. Abnormal	1.01 (0.5–1.8); *p* < 0.0001	1.80 (1.15–2.81); *p* = 0.0151
**T location** **Upper vs. middle/lower**	1.36(0.76–2.43); *p* = 0.22	1.58 (0.96–2.59); *p* = 0.081
**Liver metastasis**		
yes vs. no	0.8 (0.58–1.34); *p* = 0.56	0.95 (0.54–1.67); *p* = 0.86
**Peritoneal metastasis** yes vs. no	0.99 (0.64–1.55); *p* = 0.99	0.89 (0.56–1.40); *p* = 0.62
**Chemotherapy**		
Doublet-CT vs. Mono-CT	0.48 (0.28–0.85); *p* = 0.0010	0.49 (0.27–0.88); *p* = 0.0021
**Toxicity AE G3/G4** yes vs. no	0.99 (0.64–1.53); *p* = 0.97	0.77 (0.49–1.22); *p* = 0.29
**Drug dose reduction** yes. vs. no	0.68 (0.38–1.22); *p* = 0.14	0.60 (0.32–1.10); *p* = 0.06
**Median drugs taken** <3 vs. >3	1.22 (0.73–2.17); *p* = 0.89	0.99 (0.57–1.69); *p* = 0.75
	**Multivariate Analysis**	
	**PFS**	**OS**
**Variable**	**HR (95% CI); *p*-Value**	**HR (95% CI); *p*-Value**
**PS ECOG**		
0-1 vs. 2	6.31 (2.70–14.77); *p* < 0.0001	9.8 (3.88–24.88); *p* < 0.0001
**Albumin count**		
<3.5 vs. >3.5 g/dL	0.97 (0.52–1.79); *p* = 0.92	2.12 (1.027–4.39); *p* = 0.043
**Chemotherapy** Doublet-CT vs. Mono-CT	0.89 (0.50–1.60); *p* = 0.72	1.46 (0.67–3.19); *p* = 0.33
**G8 score** Normal vs. abnormal	2.88 (1.52–5.44); *p* = 0.0012	1.5 (0.83–2.77); *p* = 0.036.
**Weight loss** <10 vs. >10%	1.21 (0.71–2.04); *p* = 0.46	0.97 (0.58–1.62); *p* = 0.93

**Table 4 geriatrics-07-00107-t004:** Correlation between G8 score and clinicopathological features.

	G8 Normal	G8 Abnormal	*p*-Value
**Albumin count**			
>3.5 g/dL <3.5 g/dL	23 1	44 22	0.0113
**BMI**			
High Low	20 4	48 18	0.4484
**Weight-loss**			
<10% >10%	16 8	25 41	0.0288
**Liver metastasis**			
Yes No	11 13	34 32	0.8116
**Peritoneum**			
Yes No	12 12	18 48	0.0768
**Toxicity**			
Yes No	3 21	28 37	0.0148
**PS ECOG**			
0–1 2	22 2	30 36	0.0002
**Tumor site**			
GC GEC	20 4	52 14	0.8581
**Chemotherapy**			
Doublet-CT vs. Mono-CT	22 2	42 24	0.0197

## Data Availability

The related protocol was approved by Ethical Committee Fondazione Policlinico A. Gemelli—Università Cattolica del Sacro Cuore-Roma on 14 October 2020. The data presented in this study are available on request from the corresponding author. The data that support the findings of this study is not publicly available due to ethical restrictions as the information could compromise the privacy of research participants.

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
