# Peer review of "Safety and Efficacy of a First-Line Chemotherapy Tailored by G8 Score in Elderly Metastatic or Locally Advanced Gastric and Gastro-Esophageal Cancer Patients: A Real-World Analysis"

_geriatrics, 2022, doi:10.3390/geriatrics7050107_

Round 1

Reviewer 1 Report

I thank the authors for their interest in this interesting topic. I have the following questions and suggestions to improve the quality and relevance of the manuscript:

I recommend replacing the word “elderly” throughout the manuscript and instead using “older adults” as is favored throughout many geriatrics-focused journals.   

What was the reason for not performing a CGA in all patients with a score of ≤14?

G8 score is dichotomized as “high” vs “intermediate/low”. These definitions should be “normal” and “abnormal” corresponding to the G8 cut-off points.

Did all patients with a G8 score of ≤14 receive Palliative Simultaneous Care Assistance or only those with a G8 score of <12? This sentence should be clarified.

What was the recommended dose reduction at baseline? Was this different for patients with normal vs abnormal G8 score? Or for patients who received doublet vs single-agent chemotherapy?

Given that BMI and weight loss are included in the G8, was there collinearity between these variables? Please provide the correlation coefficients for table 3.

How does toxicity in this study compare to other studies of dose-reduced chemotherapy in GC/GEC? A relevant study to include in the discussion is the GO2 trial (10.1001/jamaoncol.2021.0848). Given that dose reduction was not associated with survival in this cohort, would the authors recommend an upfront dose reduction in these patients regardless of other factors?

Author Response

  1. the expression "older patients" was modified in elderly
  2. the reason for not performing a CGA in all patients with a score of ≤14 is reported in the line 233
  3. the definition off G8 score was modified in normal and abnormal
  4. all patients with G8 score of ≤14 receive Palliative Simultaneous Care Assistance as reported at line 149 - 150
  5. all patients with G8 score of ≤14 received an upfront CT dose reduction as reported at line 149 - 150
  6. A correlation between weight loss and G8 score but not with BMI  was found as reported at line 208 and in the discussion at line 246-248 suggesting a correlation between malnutrition and G8 score not with BMI which could not modified by weight loss 
  7. The GO2 trial was included in the discussion at line 268

Reviewer 2 Report

I think this is an interesting topic with a good deal of data. however, the presentation and organization are distracting to the point that it is hard to get the message across. 

I also have concerns about the design. The authors state this is a retrospective study over 7 years yet also state that each patient signed consent - did patients consent to the use of their data?

I also think there is some discrepancy in the way the G8 is described,cutoffs are described etc. There needs to be some additional attention to detail in order to keep the reader focused. 

I do think the topic is interesting yet find it hard to hang a full conclusion on the G8 as a predictor of these many outcomes -instead this could be presented as a bit more of a descriptive study. 

Author Response

  1. This is a retrospective study. The study was approved by Ethic Committee in 2020 but all patient signed an informed consent for using their clinical data for research
  2. We have added some modifications

Reviewer 3 Report

Major

1 K-M curves of PS ECOG, Albmin level, G8 should be presented.

2 In multivariate analyses results, p value of G8 is >1. It is fault.

3 Table 3 is not sorted, it is not sophisticated.

Author Response

  1. KM
  2. The data of p value of G8 score at multivariate analysis was correct
  3. the table was modified

Reviewer 4 Report

This retrospective study by Zurlo et al presents their analyses from 90 elderly patients who were treated for either gastric or gastro-esophageal cancer and assessed with the G8 score tool, in order to determine if the G8 score provides a predictor of chemotherapy (CT) safety. The study is certainly adding to our knowledge of how best to predict tolerance of the elderly population to CT, which is currently lacking. The study design is logical and they have adequately screened the selected patients. The tables indicating the patient’s characteristics, adverse events, and analyses correlating the G8 score with pathological features are appropriate. Overall, the study design is well-described and the conclusions are supported by the results, but there are some issues that should be considered. Most importantly, there are multiple grammatical issues that must be addressed, some of which are detailed below but this is by no means comprehensive. A careful grammatical review is therefore warranted.

Lines 56-57, the phrase "data resulted from retrospective does not allow to generalize...." is grammatically incorrect and difficult to understand, so recommend rewording.

Lines 58-60, the sentence is difficult to navigate, and has redundant use of "in", so suggest a re-write. Moreover, this stand-alone sentence could be combined with the previous sentence to generate a complete thought regarding chemotherapy tolerance and outcomes in the elderly, comparing monotherapy to dual (e.g., oxaliplatin and cisplatin) treatments.

Lines 61-68, the authors should carefully define CGA (as mentioned) vs. GA (assuming this is geriatric assessment), for clarity.

Lines 73 - 74, the authors describe the scoring strategy with the G8 screen with a low score as "poor" and high as "good", but then indicate that a score of 14 is abnormal, which is confusing. Why would a score approaching the high score of 17 be considered abnormal? Or did they intend to write "normal"? This needs to be addressed as presented in the mentioned publication, which indicates 14 as the threshold value for ~85% sensitivity and 65% specificity. Perhaps these values should be better described as relating to an impaired score.

Line 149 - 150, the sentence is awkward and therefore suggest rewording to provide clarity. 

Line 188, the sentence is grammatically incorrect with multiple errors, so needs editing.

Table 3 requires adjustments as some numbers are not aligned in the appropriate column.

Minor typos/suggested edits:

Line 44, add "patients" after "elderly".

Line 50, add "a" before "younger population".

Line 145, "Twenty-eight" is misspelled.

Author Response

All the suggested corrections have been done

Round 2

Reviewer 1 Report

Thank you for addressing my comments. 

Reviewer 3 Report

The authors have rivesed their manuscript along with reviewers' suggestions.